# The Role of Emotional Landmarks in Embodied and Not-Embodied Tasks

**DOI:** 10.3390/brainsci10020058

**Published:** 2020-01-21

**Authors:** Laura Piccardi, Paola Guariglia, Raffaella Nori, Massimiliano Palmiero

**Affiliations:** 1Life, Health and Environmental Science Department, L’Aquila University, 67100 L’Aquila, Italy; 2Cognitive and Motor Rehabilitation and Neuroimaging Unit, IRCCS Fondazione Santa Lucia, 00179 Rome, Italy; 3Facoltà di Scienze dell’Uomo e della Società, Università degli Studi Kore, 94100 Enna, Italy; paola.guariglia@unikore.it; 4Department of Psychology, Bologna University, 40127 Bologna, Italy; raffaella.nori@unibo.it; 5Department of Human and Social Sciences, University of Bergamo, 24129 Bergamo, Italy; massimiliano.palmiero@unibg.it

**Keywords:** landmark-based navigation, emotions, arousal, emotional cues, embodied perspective, situated cognition

## Abstract

The role of emotional landmarks in navigation has been scarcely studied. Previous findings showed that valence and arousal of landmarks increase landmark’s salience and improve performance in navigational memory tasks. However, no study has directly explored the interplay between valence and arousal of emotionally laden landmarks in embodied and not-embodied navigational tasks. At the aim, 115 college students have been subdivided in five groups according to the landmarks they were exposed (High Positive Landmarks HPL; Low Positive Landmarks LPL; High Negative Landmarks HNL; Low Negative Landmarks LNL and Neutral Landmarks NeuL). In the embodied tasks participants were asked to learn a path in a first-person perspective and to recall it after five minutes, whereas in the not-embodied tasks participants were asked to track the learned path on a silent map and to recognize landmarks among distractors. Results highlighted firstly the key role of valence in the embodied task related to the immediate learning, but not to the delayed recall of the path, probably because of the short retention interval used. Secondly, results showed the importance of the interplay between valence and arousal in the non-embodied tasks, specifically, neutral and high negative emotional landmarks yielded the lowest performance probably because of the avoidance learning effect. Implications for future research directions are discussed.

## 1. Introduction

The ability to orient in the environment is crucial for human beings. There is a wide individual difference in this ability, due to several internal (e.g., gender, familiarity; spatial skills; personal attributes) and/or external (e.g., degree of landmarks differentiation; emotional landmark; environmental layout complexity; continuous environmental changes) factors (e.g., [1,2,3,4]). In particular, human beings use landmarks, the focus of the present paper (monuments, salient objects or buildings stand out from the environment), to spatial re-orient themselves [5,6,7]. In landmark-based navigation, subjects process the exact spatial relationship between environmental objects (landmarks) and themselves. This requires aligning one’s own perspective with that of landmarks, which needs mental self-rotation [8].

Indeed, during navigation individuals process both body and external world features in accordance with the perspective taking. For example, during a first-person perspective navigation self-body information is crucial and associated with salient environmental cues (e.g., at the yellow building turn on the left), while in a world-based perspective is independent from one’s position in the environment (e.g., the church is far from the statue).

However, not all objects that individuals meet along the street are used for re-orienting; in order to be useful objects need to have structural (a prominent spatial location) [9], visual (a peculiar, shape, size and/or colour), semantic (a cultural, historical or autobiographical influences) features. Specifically, the semantic features contain a high idiosyncratic relevance since they are strictly related to personality [5].

In this vein, the key role of emotional landmarks defined in terms of valence (positive/pleasant vs. negative/unpleasant) and arousal (activating/excited vs. deactivating/calm) has only recently been investigated [3,10,11], demonstrating that positive/negative emotions increasing landmark’s salience improve performance in navigational memory tasks. More specifically, Palmiero and Piccardi [3] observed that both positive and negative emotional landmarks equally enhanced the ability to learn a path, but just positive emotional ones improved the reproduction of the path on the map. In general, these authors highlighted that emotional landmarks enhance egocentric-based topographical memory. In addition, emotionally more arousing pictures were found to produce a negative perceptual bias in estimating distances [12].

These findings are also in line with the body-specificity hypothesis advanced by Casasanto [13] in which different kinds of bodies may produce different mental representation of the space. This hypothesis refers to the theories of embodied cognition that suggest that thoughts include mental simulations or mental images of bodily experiences [14,15,16,17,18,19]. Moreover, perceptual and interactive richness landmark can alleviate cognitive load imposed on working memory by effectively embedding the learner’s cognitive activity in the environment [20]. In such a sense, it is interesting that the egocentric-based navigation is affected by emotional stimuli. Ruotolo et al. [11] showed that participants were more accurate in locating the positive landmarks along the route and drawing the route. In contrast, they found that participants judged the route as longer and were less accurate in mentally reproducing distances between landmarks when they were exposed to negative landmarks.

Taken together, these studies support the idea that spatial cognition is situated, or co-determined by both the characteristics of the body and those of the elements in the external world: Cognition is afforded and constrained by ongoing interactions between body and environment, emphasizing an intimate relationship between external artefacts and cognitive processes [21].

The novelty of the present study is to investigate both the role of valence (positive/negative) and arousal (high/low) of emotional landmarks in egocentric-based navigation. In light of Schwarz and Clore’s theory [22], which posits that positive-negative feelings are embodied information on the value of events, while activating-deactivating information contributes to provide information about the importance of the events, we hypothesized that the valence of landmarks would affect mostly embodied tasks (learning and recalling a path in a first-person perspective) rather than non-embodied tasks (tracking the learned path on a silent map and recognizing landmarks among distractors). Moreover, based on the Yerkes-Dodson law [23], according to which higher level of arousal may negatively affect memory, we also hypothesized that the high arousal of landmarks would affect both embodied and not-embodied tasks. In general, we assume that the non-embodied tasks are more prone to be affected by the interplay between valence and arousal than embodied tasks, which are supposed to be affected exclusively by valence.

## 2. Method

### 2.1. Participants

A sample 115 (54 males and 61 females; mean age: 23.61 years old; standard deviation (SD): 2.63 years old) college students from the “Department of Life, Health and Environmental Sciences”, University of L’Aquila (L’Aquila, Italy) voluntarily participated in exchange for extra credit in psychology courses. They were randomly subdivided into five groups (composed by 23 subjects per group) according to the type of landmarks they were exposed while performing the topographic memory tasks:Group with High Arousal and Positive Valence Landmarks (HPL): 23 participants (11 females and 12 males; mean age: 23.9; SD: 3.4);Group with Low Arousal and Positive Valence Landmarks (LPL): 23 participants (13 females and 10 males; mean age: 24.7; SD: 2.3);Group with High Arousal and Negative Valence Landmarks (HNL): 23 participants (12 females and 11 males; mean age: 22.3; SD: 1.4);Group with Low Arousal and Negative Valence Landmarks (LNL): 23 participants (12 females and 11 males; mean age: 24.9; SD: 2.4);Group with Neutral Landmarks (NeuL): 23 participants (13 females and 10 males mean age: 22.3; SD: 2.0).

Participants filled out the anamnesis questionnaire aimed at collecting demographic, health and alcohol/drugs assumption information. From the anamnesis, participants resulted healthy, had normal or corrected to normal (soft contact lenses or glasses) vision; they reported no neurological and/or psychiatric disorders and no problem with alcohol or drug addiction. In particular, participants self-declared no topographical orientation disorders. None of them showed the presence of navigational deficits or developmental topographical disorientation [24].

Furthermore, to exclude differences between groups with respect to the emotional intelligence, all participants underwent to the Trait Emotional Intelligence Questionnaire-Short Form (TEIQue-SF [25,26]; Italian version [27]). The TEIQue-SF is a 30-item questionnaire that investigates the global trait emotional intelligence. The univariate ANOVA showed no difference (F_4,109_ = 1.59; *p* = 0.184). In addition, all participants were submitted to PANAS [28,29] to rule out a mood manipulation due to the presence of emotional landmarks. Using Phillips et al.’s procedure [30], the individual’s mood scores at both the first and the second administration (after the completion of the Walking Corsi Test-WalCT) of the PANAS was computed by subtracting the total negative affect score from the positive affect score. Then, mood scores at the first administration (baseline) were compared with mood scores at the second administration in terms of group conditions (HPL, LPL, HNL, LNL, NeuL). Results showed no significant effect: ‘Group’ [F_(4,110)_ = 1.92, *p* = 0.11]; ‘time’ [F_(1,110)_ = 3.43, *p* = 0.067, partial eta-squared = 0.030]; interaction effect of ‘group and time’ [F_(4,110)_ = 0.71, *p* = 0.59]. These results allowed to rule out the possibility that effects on topographical memory were not due to individuals’ mood changes, but rather on emotional landmarks.

In accordance with the policy of the local ethics committee of the IRCCS Fondazione Santa Lucia (Rome, Italy) project approved in July 2017 and the Declaration of Helsinki, all participants gave their informed written consent before participating in this study.

### 2.2. Instruments

#### 2.2.1. Positive and Negative Affect Schedule (PANAS)

The Italian version [29] of PANAS [28] was administered to assess how the participant was feeling ‘right now’. It included 10 positive and 10 negative adjectives that were scored using a 5-point Likert scale, ranging from 1 (very slightly or not at all) to 5 (extremely).

#### 2.2.2. Images from The International Affective Picture System (IAPS) Inventory

Positive/negative valence with high/low arousal and neutral landmarks were selected from the Images from The International Affective Picture System (IAPS) Inventory [31,32]. Specifically, 15 affect-laden images were printed in the size 30 cm × 30 cm. Pictures include standardized coloured photographs representing three categories of emotional stimuli (positive, negative and neutral), being scored in terms of valence (ranging from pleasant to unpleasant) and arousal (ranging from high to low). In this study, images were differentiated according to their valence and arousal, which are the two fundamental aspects of emotionality [33], according to the original scores [32], as follows:−3 positive emotional images with pleasant valence (mean = 8.02, (SD) = 0.25) and high arousal (mean = 6.21, SD = 0.47), namely the images of ‘beach’, ‘skier’ and ‘sailing’;−3 positive emotional images with pleasant valence (mean = 7.42, SD = 0.35) and low arousal (mean = 3.11; SD = 0.13), namely the images of ‘rabbit’, ‘flower’, and ‘clouds’;−3 negative emotional images with unpleasant valence (mean = 1.55, SD = 0.12) and high arousal (mean = 6.82, SD = 0.6), namely the images of ‘face mutilated’, ‘soldier’, and ‘dog’;−3 negative emotional images with unpleasant valence (mean = 3.37, SD = 0.44) and low arousal (mean = 3.93, SD = 0.06), namely the images of ‘homeless man’, ‘exhaust’, and ‘woman’;−3 neutral emotional images both in terms of valence (mean = 6.03, SD = 1.18) and arousal (mean = 3.37, SD = 0.22), namely the images of ‘parrots’, ‘cow’, and ‘man’.

Neutral emotional images were selected following valence and arousal values indicated in Nowicka et al.’s study [34].

#### 2.2.3. The Walking Corsi Test (WalCT)

The Walking Corsi Test (WalCT) [35,36] was used to measure topographic working memory. It consists of nine squares (30 cm × 30 cm) placed in an empty room within a walking space (3 m × 2.5 m) reproducing the well-known Corsi Block Tapping Test (1:10 scale) [37]. As in Piccardi et al. [38], three 30 × 30 cm pictures of landmarks were placed on three out of the nine WalCT squares (see Figure 1). However, in the present study we used 15 affect-laden images previously selected from the IAPS Inventory [31,32] according to the high/low arousal and the positive/negative valence. Emotional or neutral landmarks were placed on the squares of the WalCT, in the same position for all conditions for having comparable intersections among squares and the same spatial distances with respect to the path to learn. (See Figure 1).

### 2.3. Embodied Tasks

#### 2.3.1. Learning of the Path

The examiner showed an eight-square path (the same for all conditions) by walking on squares at a rate of one square per 2 s. Participants were asked to learn the path demonstrated by the examiner. When participants were able to reproduce correctly the path three times in a row, the examiner considered that the learning criterion had been reached. Anyway, if the subject was not able to repeat the path, the examiner showed it to him/her for a maximum of 18 repetitions until the learning criterion or the maximum number of repetition were reached. The learning score was calculated as follows: one point was given for each square correctly walked within the right order of the sequence, until learning has taken place; then, eight points were summed for each of the remaining trials (up to the 18th repetition; maximum score = 144).

#### 2.3.2. Delayed Recall of the Path

Five minutes later, participants were asked to reproduce again the path by walking the previously learned eight-square path. The delayed recall score was obtained by summing the number of squares correctly walked (maximum score = 8).

### 2.4. Not-Embodied Tasks

#### 2.4.1. Drawing of the Path

Examiner asked participants to retrace with a felt-tip pen the eight-square path on the silent map representing the WalCT. The silent map score was the number of squares correctly drawn (maximum score = 8).

#### 2.4.2. Recognition of Landmarks

Participants were also asked to recognize the emotional or neutral landmarks used among 3 distractors comparable in terms of valence and arousal (for emotional images with pleasant valence and high arousal, valence: mean = 8.05, SD = 0.25, arousal: mean = 6.11, SD = 0.52; for emotional images with pleasant valence and low arousal, valence: mean = 7.46, SD = 0.51, arousal: mean = 3.27, SD = 0.05; for emotional images with unpleasant valence and high arousal, valence: mean = 1.71, SD = 0.17, arousal: mean = 7.11, SD = 0.21; for emotional images with unpleasant valence and low arousal, valence: mean = 3.35, SD = 0.11, arousal: mean = 3.8, SD = 0.26; for neutral images, valence: mean = 5.95, SD = 0.55, arousal: mean = 3.43, SD = 0.33).

### 2.5. Experimental Procedure

At the beginning, participants were given brief instructions to participate in to the experiment, they provided their written informed consent and they were randomly assigned to one of the five experimental groups (HPL, LPL, HNL, LNL and NeuL). Then, they filled in the PANAS for the first time. Afterwards, participants performed the three topographical memory tasks (learning, delayed recall and the silent map of the eight-square path) in all conditions respectively. Afterwards, participants filled in the PANAS for the second time. Finally, they performed the recognition landmark task.

## 3. Results

A significant threshold was set at *p* = 0.05/4 = 0.0125 by using Bonferroni’s correction for multiple comparisons to avoid to commit a Type 1 error.

### 3.1. Analyses on Topographical Memory

In order to exclude effects on learning, delayed recall and reproduction of the eight-square sequence due to gender, firstly three separate ANOVAs were performed with gender as independent variable. Results revealed no significant gender difference in learning [F_(1,113)_ = 3.62, *p* = 0.06]; delayed recall [F_(1,113)_ = 0.05, *p* = 0.83] and reproduction [F_(1,113)_ = 0.001, *p* = 0.98] scores. Given that gender produced no effect on the variables of interest, subsequent analyses were carried without considering gender differences.

### 3.2. First Embodied Task—Learning of the Path

The Univariate ANOVA carried out on the learning score of the sequence revealed an effect of group [F_(4,110)_ = 4.66, *p* = 0.002, partial eta-squared = 0.145; observed power = 0.94]: Post hoc analysis (LSD: *p* < 0.05) showed that all groups exposed to emotional landmarks, regardless of the valence and the arousal performed better that the neutral group; no difference was found between the HPL, HNL, LPL and LNL.

### 3.3. Second Embodied Task—Delayed Recall of the Path

The Univariate ANOVA carried out on the delayed recall score of the sequence revealed no group difference [F_(4,110)_ = 0.58, *p* = 0.68].

### 3.4. First Non-Embodied Task—Drawing of the Path

The Univariate ANOVA carried out on the learning score of the sequence revealed an effect of group [F_(4,110)_ = 7.14, *p* = 0.00005, partial eta-squared = 0.206; observed power = 0.99]: Post hoc analysis (LSD: *p* < 0.05) showed that the HPL, LPL and LNL scored higher than the neutral group. Only the HNL scored as the neutral group and lower than the other groups. No difference was found between HPL and LPL and between LPL and LNL.

### 3.5. Second Non-Embodied Task—Recognition of Landmarks

Finally, the Univariate ANOVA carried out on the recognition landmark score revealed an effect of group [F_(4,110)_ = 12.71, *p* = 0.00005, partial eta-squared = 0.316; observed power = 0.99]. Post hoc analysis (LSD: *p* < 0.05) showed that the NeuL group had a better performance than the two groups with high arousal (HPL and HNL), and no difference between the two groups with low arousal (LPL and LNL). The HPL, scored lower than all other groups (see Figure 2).

## 4. Discussion

In the present study we took into consideration both valence and arousal of emotional landmarks, hypothesizing that positive/negative valence and high/low arousal may produce effects on the navigational task (embodied or not-embodied) that participants had to perform. Considering that valence is more embodied than arousal, we expected that learning in first-person perspective (embodied tasks) and recalling a path may be more affected by positive/negative landmarks regardless of their arousal. Anyway, it is also known that higher arousal may negatively affect memory [23]. In this vein, we also expected to find significant differences due to high/low-arousal emotional landmarks in both embodied and non-embodied tasks.

The results are partially in line with our initial hypotheses, given that although participants exposed to positive or negative landmarks were faster in learning the path in a first-person perspective regardless by the high/low arousal of the landmark, no differences due to landmarks were found in the delayed recall task. In addition, participants exposed to negative/high arousal landmarks were worse than the other three emotional landmark groups only in the drawing of the path task. Results confirmed and extended Palmiero and Piccardi’s [3] study, showing that both positive and negative landmark groups (with high arousal) enhanced the learning of the path task, with no advantages due to emotional landmarks in the delayed recall phase, whereas only the positive landmark group was facilitated in the reproduction of the path on the map.

Thus, on the one hand, in accordance with the first hypothesis, in the present study, we found that the valence, regardless of the arousal, supports the embodied immediate learning, but not the delayed recall of the path. The lack of no effect of landmarks on the delayed recall task could be related to the retention interval. In fact, we asked participants to repeat the 8-squares path after 5 min. Carpenter et al. [39] demonstrated that it is more likely to yield a ceiling effect at the 5-min retention interval with respect to longer intervals (i.e., 30 min, 1 day, 2 days, 7 days, 14 days, or 42 days). Therefore, in future studies it would be interesting to vary the retention interval introducing longer intervals of time to observe a possible effect of the emotional landmarks even on the delayed recall phase.

On the other hand, in accordance with the second hypothesis, we found that in not-embodied task, when the task request is less situated, the interplay between valence and arousal is crucial. Indeed, in tracking the route on the map participants with high/low positive emotional landmarks and low negative emotional landmarks were more accurate than neutral landmarks and high negative emotional landmarks. These findings are in line with Ruotolo et al.’s results [11], revealing that participants in the positive condition were more accurate than those in the neutral condition in drawing the route as well as in indicating the landmarks location. In addition, Palmiero and Piccardi [3] highlighted that emotional landmarks, especially positive, promote accurate spatial representation. The fact that only low negative landmarks and not high negative landmarks increased the accuracy may be explained within the avoidance learning. Avoidance of genuinely threatening situations is a key characteristic of adaptive fear (for a review see, [40]): People avoid to enter in a building after a major earthquake, avoid to approach a snake, etc. Thus, in a not-embodied task participants have to access to an abstract representation of the path learned and they are supported by high/low positive emotions and not by those that are too negative or neutral.

One might argue that the one of the not-embodied task, the drawing task, could be a partially embodied task as the participants could mentally simulate their previous exploration in order to draw it. However, to draw the path on the map requires the transformation of the egocentric representation of the environment in which the participant has to navigate through in an allocentric spatial not-embodied representation that mitigates this criticism.

Another possible explanation could be related to the cognitive effort necessary to carry on this task. Positive emotions may support the attentional mechanisms devoted to the task favouring the mental transformation of egocentric information in allocentric one. According to Ashby et al. [41], the positive mood should increase biological mechanisms of increasing of dopamine favouring cognitive processes, such as working memory, and as a consequence facilitating the execution of more complex tasks, such as building a map-like representation. Although we did not find a mood manipulation as demonstrated by the performance on PANAS, we could conceive that just emotional landmarks may elicit these mechanisms. Indeed, the emotional arousal per se may affect the consolidation of long-term memory through the interaction of amygdala with other brain regions [42,43]. According to Scheibe and Carstensen [44], neural mechanisms underlying the memory of low and high arousal stimuli are different. Generally speaking, low-arousal stimuli require emotional regulation and more controlled processing performed by prefrontal brain regions, while high-arousal stimuli are more automatic and associated with functions of the amygdala [45,46,47]. Moreover, in the recognition of landmarks, we found that high-arousal stimuli were recognized worse than low-arousal stimuli and neutral stimuli, as if emotions worsened the ability to recognize landmarks out of context. Considering that environmental representation is situated, both the individual and the external features of the world are important in the memory process, when landmarks are shown separately from the context the advantages of emotionally laden landmarks disappears and the importance to remember high positive/negative landmarks has no evolutionary advantage.

## 5. Conclusions

In conclusion, the present study sheds some light on the interplay between emotions and topographic learning highlighting the importance of valence but not of arousal in building up an embodied environmental map. Conversely, when individuals access to a spatial representation regardless of their environmental position the interaction between arousal and valence is crucial, as well as in the landmark recognition in which neither the individual’s position nor the environment is required to solve the task. Present findings may have effects on the training devoted to rehabilitate patients suffering from amnesia or memory disorders and patients affected by anterograde topographic disorientation. Considering that when the retention interval is short, the most sensitive phase is the encoding/learning phase [35,39,48,49,50], introducing emotional cues during the learning phase of navigational tasks may help not only healthy individuals, but also patients in rehabilitation. In this direction, seminal studies [51,52] demonstrated the interaction between emotions and memory in other memory domains (episodic and autobiography memories), showing that memories for neutral events decrease over time if compared with arousing events. In other words, emotional stimuli, arranged in terms of valence and arousal might play a key role in facilitating both embodied and non-embodied tasks in different types of population.

## Figures and Tables

**Figure 1 brainsci-10-00058-f001:**
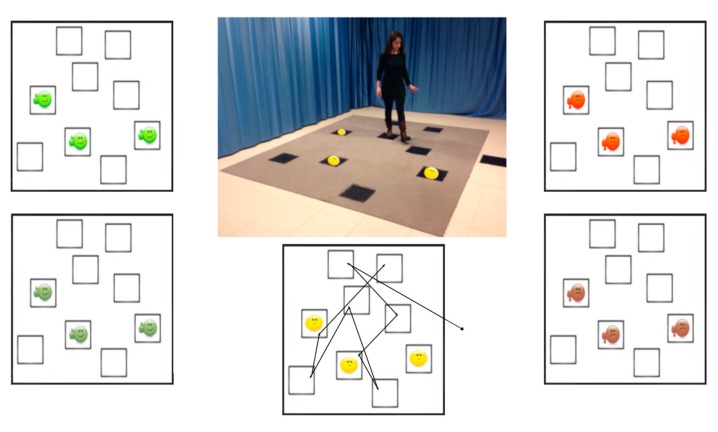
The landmark-based navigational memory task (the Walking Corsi Test with emotional landmarks). The eight-square path was designed in order to let participants move through the squares, as showed by the red line. In the centre of the figure is displayed the experimental set-up with a participant who performed the learning path task. Written informed consent was obtained from the person represented in the picture for publication. A copy of the written consent is available for review by the Editor-in-Chief of this journal. Around the centre figure are shown the disposition of the high/low arousal positive (on the left), high/low arousal negative (on the right) and neutral (below and in the photo) landmarks through the path.

**Figure 2 brainsci-10-00058-f002:**
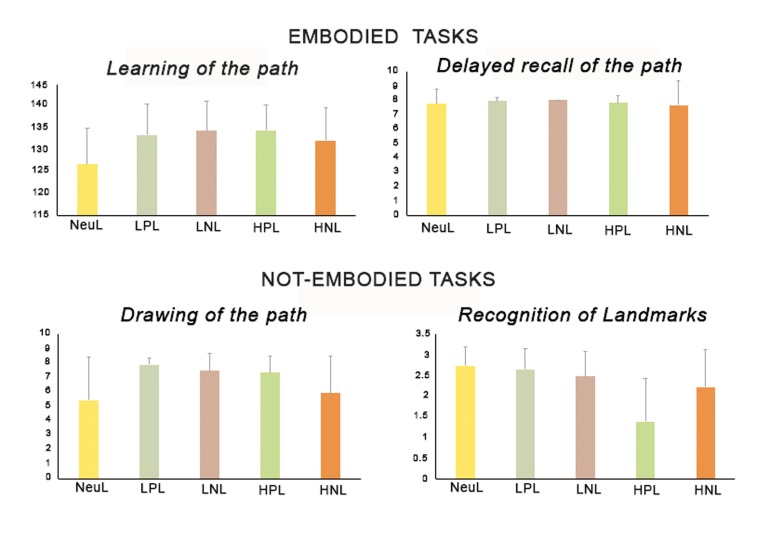
Mean performances and SD of participants in embodied and not-embodied tasks.

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
