# Peer review of "The Role of Emotional Landmarks in Embodied and Not-Embodied Tasks"

_brainsci, 2020, doi:10.3390/brainsci10020058_

Round 1
Reviewer 1 Report
The aim of this research work was to understand the role of landmarks' valence and arousal values in navigation. For this purpose, 5 groups of participants carried out the Walking version of the Corsi task. The five groups learned a path characterized by different images, i.e. with high or low arousal, and with positive or negative valence; a neutral condition was also considered. Specifically, participants were asked to perform two embodied tasks, i.e. learning the route and recall the route, and two non-embodied tasks, such as drawing the route and landmarks recognition. The results showed that groups with emotionally-laden images had a better learning than groups with neutral elements, however no advantage of the emotional conditions was observed in the recall task. In addition, subjects with negative image and high arousal were worse in drawing the route.
General comment: The work is interesting and well written. It adds evidence about the role of the emotionally laden landmarks in navigation. However, there are some points I would like the authors address.
1) The authors make a distinction between embodied and non-embodied tasks, however the route drawing task could be an embodied task as the participants could mentally simulate their previous exploration in order to draw it. In short, it is not clear how the distinction between embodied and non-embodied tasks can be theoretically and operationally justified.
2) The authors expect an interaction between arousal and valence in non-embodied tasks, however, the analyses conducted would not allow to speak of an interaction effect. Would it not have been more appropriate to conduct a 2X2 ANOVA with the four emotional conditions?
3) Since several measures were present, why wasn't a MANOVA carried out? Furthermore, considering the different ANOVAs performed, how was the problem of type I error addressed?
4) How was the number of participants per condition decided? Please specify
Author Response
General comment: The work is interesting and well written. It adds evidence about the role of the emotionally laden landmarks in navigation. However, there are some points I would like the authors address.
1) The authors make a distinction between embodied and non-embodied tasks, however the route drawing task could be an embodied task as the participants could mentally simulate their previous exploration in order to draw it. In short, it is not clear how the distinction between embodied and non-embodied tasks can be theoretically and operationally justified.
Reply: We agree with the Reviewer that the participant could mentally simulate their previous embodied navigation however to draw the path on the map they have to transformate the egocentric representation of the path in an allocentric one. For such a reason we believe that this task is not-embodied as landmark recognition is. We now add a sentence in the Discussion about this task and its properties.
2) The authors expect an interaction between arousal and valence in non-embodied tasks, however, the analyses conducted would not allow to speak of an interaction effect. Would it not have been more appropriate to conduct a 2X2 ANOVA with the four emotional conditions?
Reply: Following Reviewer’s suggestion we understand that speaking about interaction may be misleading, for such a reason we modified appropriately in the text.
3) Since several measures were present, why wasn't a MANOVA carried out?
Reply: We considered tasks very different from each other, so we retained more statistically correct to make separate analyses. In general in the most of previous studies using Walking Corsi Test separate ANOVAs have been carried out for learning and delayed recall, so we adopted the same model of astatistical analysis.
Furthermore, considering the different ANOVAs performed, how was the problem of type I error addressed?
Reply: To check for the type I error, we used the Boferroni’s correction for multiple comparison taking into account for a smaller p value, results do not change and we now added the following sentence at the beginning of the results: A significant threshold was set at p = 0.05/4 = 0.0125 by using Bonferroni’s correction for multiple comparisons.
4) How was the number of participants per condition decided? Please specify
Reply: In absence of a G-Power performed pre sample selection, we computed the observed power and added them in the Result section. Taking into consideration that observed power is high we believe that the number of participants per group is enough.

Reviewer 2 Report
The manuscript entitled “The role of emotional landmarks in embodied and not-embodied tasks” aimed at investigating the impact of the interaction between valence and arousal of emotionally laden landmarks in embodied and not-embodied navigational tasks. One-hundred-and-fifteen students were exposed to five different combination of valence and arousal landmarks in embodied and not-embodied tasks. Results showed that valence had a significant impact in the embodied task related to the immediate but not to the delayed learning, and the impact of the interaction between valence and arousal in the non-embodied tasks, in which neutral and high negative emotional landmarks produced the lowest performance. Authors discussed theoretical and practical implications of results obtained.
I carefully read the manuscript, and I think it would be of interest for the readers of Brain Sciences. The main strength of the present study is in exploring spatial cognition in a situated perspective, determined by the interplay of internal/cognitive and external/environmental processes. Indeed, there are nowadays several studies assessing the impact of different types of landmarks on spatial orientation and navigation abilities, but little investigation had been conducted on the role of emotional landmarks in an egocentric frame of reference.
The research question of this study is well defined and represent a novelty and a promising step in the field of situated spatial cognition. Methods section is clear and its contents are well-organized. Results are appropriately reported and interpreted; statistics used are pertinent with the aims of the study. The paper is well written and the English-language used is clear. Figures are understandable and well laid out. Discussion follows a commendable logic which elaborates the rationale explained in the Introduction.
Author Response
Authors would thank Reviewer 2 for his/her positive comments on their manuscript.